# Understanding university students' experiences of sitting while studying at home: A qualitative study

Hannah Clare Wood[1]*, Sanjana Prabhakar[1], Rebecca Upsher[1], Myanna Duncan[1], Eleanor J. Dommett[1], Benjamin Gardner[2]

**1** Department of Psychology, Institute of Psychiatry, Psychology & Neuroscience, King's College London, London, United Kingdom, **2** Department of Psychology, University of Surrey, Guildford, United Kingdom

* hannah.c.wood@kcl.ac.uk

## Abstract

University students are typically highly sedentary, which is associated with adverse physical and mental health outcomes. Attempts to understand university students' sedentary behaviour have typically focused on on-campus teaching and learning activities. While such research has documented that students perceive studying as one of the main barriers to reducing sedentary behaviour, there is little understanding of how and why studying incurs sitting, especially during home-based studying. We investigated how students' experience sitting when studying at home. Fourteen UK undergraduates (10 female; mean age = 20 years) participated in semi-structured interviews that explored their experiences, beliefs, and attitudes regarding sitting while studying at home. Data were analysed using Reflexive Thematic Analysis. Four themes were constructed, focusing on knowledge and attitudes towards sitting, prioritisation of study tasks, sitting as an integral part of the study environment, and sitting as part of the optimal mental 'zone'. Whilst participants demonstrated awareness of the potential health risks associated with prolonged sitting they prioritised productivity when studying at home and believed that interrupting their sitting would compromise productivity, making home-based studying particularly conducive to sedentary behaviour. These findings suggest that intervention designers should more explicitly consider the home environment when aiming to reduce student sedentary behaviour and should seek to reduce sitting in a way that promotes, or at least does not interfere with, productivity.

**Data Availability Statement:** All anonymised interview transcripts are available on the OSF (https://osf.io/6rdvf/?view_only=9b901d9f4f504dd3877dc3f9dbc0848d).

## Introduction

Sedentary behaviour–i.e., any waking behaviour that uses low levels of energy expenditure while in a sitting, reclining, or lying posture–is associated with adverse physical and mental health outcomes, largely independently of engagement in physical activity [1–3]. Greater sedentary time has been linked to greater risk of cardiovascular disease, type 2 diabetes, all-cause mortality [2, 4, 5], increased depression, and anxiety [6, 7]. In university students, levels of sedentary behaviour have also been positively associated with stress, anxiety, depression, and

**Funding:** This work was supported by the Economic and Social Research Council: HW is in receipt of PhD funding from the London Interdisciplinary Social Science Doctoral Training Partnership [grant number: ES/P000703/1], https://liss-dtp.ac.uk. There was no additional external or internal funding received for this study.

**Competing interests:** The authors have declared that no competing interests exist.

suicidal ideation, independent of the effects of physical activity [8, 9]. Yet, university students typically accumulate many hours of seated activities, more so than young adults not in university education [10, 11]. One review of 23 studies, conducted across multiple countries, found that students spent an average of 10.69 hours a day without moving [12]. Calls have been made for changes to university culture and policy to encourage movement throughout the university experience [13–15].

Reducing sitting among students requires understanding why students sit. Sitting is often incidentally incurred by engagement in activities that necessitate sitting (e.g., attending a lecture in an all-seated classroom) rather than resulting from a conscious decision to sit [16]. Students are likely particularly sedentary because many teaching and learning activities, such as attending lectures or seminars, are typically conducted while sitting [17, 18]. Qualitative research suggests that students perceive studying and their academic workload to be among the main barriers to reducing sedentary behaviour [17, 19, 20]. However, such work has only documented that studying seems to encourage sitting, not how or why studying incurs sedentary behaviour, and therefore how such behaviour might be addressed. Equally, while research into teaching and learning activities conducive to sitting has tended to focus on in-person teaching [21], students also spend significant amounts of time studying privately. Home-based study has become particularly important since the COVID-19 pandemic, which led to a rise in the adoption of blended learning approaches to teaching delivery, which involve more online activities and less on-campus teaching time [22–24]. Little evidence is available regarding how students undertake teaching and learning activities when studying from home (e.g., online lectures), or their experiences of sitting as part of these activities (however, see [25]). Analogous research among office workers has suggested that the way in which work tasks are adapted for performance at home inhibits physical activity and facilitates sedentary behaviour compared to doing the same tasks in the office [26, 27].

Studying at home forms a significant part of student life, yet little is currently known about how students study at home and how their at-home study practices might relate to sitting. Developing this understanding is needed to inform the development of future interventions to reduce students' study-related sitting. The present study aimed to develop understanding of how university students study at home and their beliefs, attitudes, and experiences regarding home-based studying and sedentary behaviour.

## Method

### Participants and procedure

We sought to recruit full-time undergraduate students at a UK university, aged 18 years or above, with experience of studying at home. Prior to engaging in the present study, participants were recruited to a separate survey study about sedentary behaviour, via advertisements posted on social media channels and two universities' research participation recruitment platforms. The advertisements contained a URL to an online survey, via which participants self-declared their eligibility, reported demographic information and sitting behaviour, and, for the purpose of a separate study not reported here, their capability, opportunity, and motivation to reduce their sitting. Those who completed the survey were invited to volunteer for this study by providing their university email address, which we then used to arrange an interview for the present study. The demographic information collected via the survey was used to describe the sample for this study. Written informed consent was obtained via email for all participants.

Participants were recruited between 11th February- 28th March 2022, in the 2021–22 academic year, with interviews taking place within a week of participants signing up for the study.

**Table 1. Participant demographic information.**

|  |  | n | % |
|---|---|---|---|
| Gender | Female | 10 | 71 |
|  | Male | 4 | 29 |
| Year of study | Third/final | 8 | 57 |
|  | First | 4 | 29 |
|  | Second | 2 | 14 |
| Ethnicity | Asian or Asian British | 6 | 43 |
|  | Arab | 4 | 29 |
|  | White or White British | 3 | 21 |
|  | Not stated | 1 | 7 |
| Subject area | Psychology | 3 | 21 |
|  | Law | 3 | 21 |
|  | Medicine and allied subjects | 3 | 21 |
|  | Biological and sport sciences | 2 | 14 |
|  | Business management | 1 | 7 |
|  | Engineering | 1 | 7 |
|  | Media | 1 | 7 |

At all UK universities, the preceding two academic years had involved studying being moved unexpectedly and exclusively online in response to COVID-19 lockdowns (March 2020-Aug 2020), and then being transitioned, at least in part, to blended learning delivery (Sept 2020–2021). Fourteen undergraduate students were interviewed (mean age = 20.14, SD = 0.95; range = 19–22 years). Sample size was determined pragmatically within the time constraints of the study, but the resulting data set was deemed sufficiently rich and diverse to achieve data saturation. Most were female, in their final-year, and studying psychology or a health-related subject (see Table 1). Data on participants sitting time were collected as part of the online survey and is included here to provide additional context and characterisation of the sample. In response to a total and domain-specific sitting questionnaire ([28], modified for a student population), participants reported spending an average of 15 hours 38 mins a day (SD = 5 hours 51 mins) sitting, including 6 hours a day (SD = 2 hours 24 min) of sitting during studying, on weekdays. For weekend days, participants reported an average of 13 hours 18 mins a day (SD = 4 hours 41 mins) sitting, including 3 hours 15 mins (SD = 1 hour 58 mins) of study-specific sitting (n = 13; missing data for one participant).

Semi-structured interviews were conducted by one of five undergraduate psychology students (three female, one non-binary, and one male; including SP), under the supervision and guidance of senior qualitative researchers (BG, RU). Interviewers had no prior personal relationship with participants. All interviewers had received training in qualitative interviewing as part of their studies, as well as additional project-specific training from the supervisory team (BG, RU).

Interviews were conducted online using Microsoft Teams and were audio-recorded and auto-transcribed. Only the interviewer and the participants were present for the interview, which ranged between 30–60 minutes in duration. After the interview, participants were verbally debriefed and sent a follow-up email that included links to information about mental health and wellbeing services. Participants received a £10 Amazon voucher upon completion of the study. All procedures were approved by the King's College London Research Ethics Committee (reference: LRS/DP-21/22-26926).

### Interview schedule

The semi-structured interview guide (see S1 File) was developed collaboratively, based on the senior researcher's knowledge of the area and the student interviewers' personal experiences of studying at home. Topics covered by the interview guide included: studying at home practices and priorities (e.g., 'can you describe what a typical at-home study session looks like for you?'); sitting time and awareness of health implications (e.g., 'how much time would you say you spend sitting during a typical session of studying at home?'); questions regarding participants' responses to survey items (e.g., 'we asked you about whether you have social opportunities to break up and limit your sitting time, you responded 'X'. Can you explain this in more detail for me?') and potential acceptability of possible intervention ideas (e.g., 'what would help you personally to take more breaks from sitting while studying at home?'). The guide was developed and refined over an iterative period of pilot testing with the research team and a practice interview conducted with a student (excluded from the final dataset).

### Analysis

Automated transcripts were checked for accuracy and edited where necessary by the corresponding interviewer. Data were analysed using inductive Reflexive Thematic Analysis [29], underpinned by critical realist assumptions, using NVivo 14 [30]. Analysis began with a period of familiarisation with the transcripts, during which initial ideas were noted down. Next, systematic coding was undertaken of all potentially relevant extracts in relation to the research question. Preliminary codes were physically clustered into potential themes through an iterative process of arranging paper copies to visually represent the relationships between the developing themes, and then comparing this potential interpretation of the data to the data extracts the codes represented. Initially constructed codes and clusters of codes (i.e., themes) were iteratively refined as analysis progressed. When a coherent and authentic set of themes was achieved, the themes' central organising concepts and definitions were refined. Finally, theme names were finalised, including short data extracts to ensure and demonstrate grounding of theme labels in the data, and illustrative data extracts were selected to represent each theme.

Analysis was primarily conducted by HW, who met frequently with BG, a senior qualitative researcher, who reviewed the developing themes, and provided 'critical friend' feedback on coding decisions, interpretation, and theme generation [31]. Discussions led to further refinement and reorganisation of themes throughout the analysis process, resulting in the final set of themes, which were confirmed by BG to offer a credible interpretation of the data.

### Researcher positionality

The study was conceived by BG and RU to investigate students' studying at home practices in relation to health and wellbeing, as a progression from doctoral research supervised by BG and MD into health behaviour and wellbeing among normally office-based workers who worked from home during the 2020 UK COVID-19 lockdown [26]. Data collection was carried out in 2022 by undergraduate students as part of their final-year research project, under direct supervision of BG and RU. In 2023, BG invited HW to analyse the dataset to inform HW's PhD project, which sought to develop an intervention to increase student physical activity within university education. HW is a PhD student—supervised by BG, MD, and EJD—who frequently works from home and incurs considerable sitting time in doing so, but actively tries to break this up through short breaks and use of a standing desk.

## Results

Four themes were constructed, focusing on knowledge and attitudes towards sitting, prioritisation of study tasks, sitting as an integral part of the optimal study environment, and finding the optimal mental 'zone'.

### 'Sitting's not good': Knowledge and attitudes towards sitting

Participants reported that, while studying, they typically spend long of time sitting with limited break-taking. Many participants felt that they sit for '*too long*' (Participant 17; P17) when studying at home and demonstrated awareness that extended sitting could compromise health in some way ('*I guess that sitting for three hours is probably extremely unhealthy*'; P6). Some participants described having negative affective responses to prolonged sitting, or to their reflections on prolonged sitting:

> *sometimes even if you are studying and accomplishing a lot it does feel bad that you're sitting down . . . even when you're producing work, there's always that thought in your head that you're just sitting down. You're not really moving so you're not doing anything basically* (P10).

Overall, however, participants appeared to have limited knowledge around specific risks of extended sitting. When asked about the potential consequences of sedentary behaviour, most expressed uncertainty ('*I don't know much apart from stuff like oh, it's just not good for your spine, that you should get up a little bit*', P8), or cited short-term physical consequences ('*I think it's bad for blood circulation, I think that's about it, probably your posture too*' [P10]). None mentioned any longer-term health implications of sedentary behaviour.

Despite a general awareness that extended sitting is '*not good*' (P14), participants did not appear to sufficiently value taking regular breaks from sitting, with most reporting extended periods of sitting while studying. Indeed, the word 'breaks' was often used to describe momentary departures from work tasks, rather than breaking up sitting, and several participants reported that they often remained seated while 'taking a break':

> *maybe I'll watch a YouTube video but that means I'm still sitting down after all so I don't really stand up unless there's a physical need to* (P6).

Many described becoming aware of having been sitting for 'too long' only when experiencing physical discomfort ('*when I feel my legs start to feel numb then I will realise that I've been sitting for way too long*' [P14]; '*I don't think I pay attention to it unless it does get like an extreme length of time, you know, like four or five hours*' [P12]). When presented with the suggestion that they might monitor their sitting time, several stated that monitoring work and break times would be too burdensome given the cognitively demanding nature of studying.

### 'If I'm studying, I'm studying': Prioritisation of study tasks

Participants assigned a high value to productivity and described positive affective responses to having achieved their study productivity goals ('*I feel fulfilled in a way where it's like, okay, I'm satisfied, it's like that internal self-satisfaction . . . I know I've done a lot today*'; P8). Importantly, participants appeared to prioritise the completion of study tasks when studying at home and viewed spending prolonged time studying–and sitting–as necessary to maximise their productivity. Several participants felt that efficiency was important to their studying practices, and

sought to minimise both the time spent studying ('*when I watch a lecture I watch it at double speed*' [P5]) and the effort required to meet productivity goals:

> *everyone just wants to have a way of studying that works best for them and requires the least effort in a way . . . so they're not having to really, really, really work [hard] [. . . to achieve] something small* (P12).

Many participants viewed time away from work as wasted time because it compromised their efficiency ('*I'll make sure I'm not already hydrated so I don't need to go to the toilet*' [P2]). Many reported ensuring that they had everything required to study effectively available at the start of a study session to avoid having to interrupt studying. Activities like commuting, for example, were viewed as an unnecessary loss of time that could otherwise be spent studying:

> *[if I wanted to study on campus] I'd have to walk about a mile or two to get to uni buildings so [studying at home] cuts out the commuting time. It's about a half-hour walk in total [so] that's an hour of lost productivity* (P4).

Breaks from sitting were also typically portrayed as an unnecessary disruption of studying ('*if I'm taking a break, I'm not studying*'; P5). Indeed, many participants viewed 'good' studying as the ability to keep going without breaks ('*a good day of studying [involves] . . . not having to get up and move . . . no breaks, just power through*'; P10). Some participants alluded to a social stigma around taking breaks from studying, such that seeing others taking breaks alleviated potential guilt about their own break-taking ('*when I see other people in my course who are also taking breaks, and not just studying all the time, it does make me feel better about taking breaks*' [P3]). Several participants felt that they had to earn their breaks, by first engaging in periods of productive study:

> *if I finish the lectures for one module for that week then I normally stand up . . . because at that point it's been like 3 hours work, I feel like I've probably completed it so I can get up and do something* (P12).

When asked about the possibility that they might alter their study practices to incorporate more breaks from sitting, participants generally expressed reluctance to change, because their years of experience had given them confidence in knowing 'what works' for them:

> *I like the way I study, I feel like, I get this [work] out and then don't get out until I've done my work, so I don't feel like taking breaks when studying is the best course of action for me personally* (P4).

Participants viewed breaks from sitting more positively where they used them strategically to boost their productivity:

> *I know myself well enough that even as much as I want to just continue and get my work done, it's going to be counterproductive if I just keep pushing myself and I know it's going to be more productive . . . to take a break [from sitting at my desk] because it's just going to do more harm than good if I just try and study more* (P8).

Participants who held this view described using breaks to '*refresh your mind*' (P10) and felt that, after taking a break from sitting, they could '*refocus better*' (P14) and sustain their

concentration and energy for longer. Some reported breaking studying into time-specific chunks, which enabled them to schedule regular breaks while also achieving their productivity goals:

> *I do it in half an hour blocks because after that either my eyes get tired or I get tired . . . so I like to do it in half an hour blocks and then take a break for like 5 to 10 minutes, so I'm not getting too distracted, I've had enough time sort of walk round . . . and then come back and finish the next half an hour block* (P1).

## 'I have to put my phone away': Sitting as an integral part of the optimal study environment

Participants described the importance of having what they felt was the 'right' physical environment in which to study productively. Participants reported striving to create this environment by, for example, tidying their desk, and attempting to shield this 'perfect' environment by minimising potential distractions and interruptions. Actions taken to minimise distractions included telling household members that they should not be disturbed and moving non-study devices out of sight ('*I always have to put my phone away otherwise it's game over*' [P5]).

Maintaining this optimal environment was viewed as particularly important to avoid potential temptations to engage in other activities ('*there's always other things I could do like fold the laundry, like make a snack, call someone, and there's not any limitations to doing other things like that*' [P11]). To resist such temptations, most participants studied alone in their bedrooms and sat at a desk. Sitting was thus seen as integral to effective studying:

> *Sure, I don't want to sit down for very long periods of time, so it's not good, but . . . I feel like when I do sit down it means that I'm trying to do work I guess and so it's like I would rather be doing work then being distracted and doing something else and not sitting down* (P7).

Breaks from sitting were frequently viewed as distracting, not only because it diminished work time, but also because moving away from the desk exposed them to further distractions:

> *if I forget something then I'm going to have to get up and I know that if I keep getting up then I'll probably find something to distract myself with and . . . defeat the point of going to study* (P1).

Thus, many participants felt that standing during studying had to serve a purpose ('*when I'm studying I wouldn't get up unless I have to*' [P8]), such as to make lunch or answer a doorbell.

Some participants described the physical environment at home as a constraint on movement:

> *there is no space . . . so even when you take a break, like in an apartment, sometimes you'd go to a kitchen it's at least after a couple of steps. Mine is literally two steps up, like far from my desk so it's a very limited space to live in. So I feel like there are no opportunities . . . for me to actually move around so I don't do it* (P10).

Several stated that they would occasionally vary their physical study environment, often to serve the additional goal of enhanced physical comfort alongside their goal of productivity,

and so study from bed ('*in the morning I usually stay at my desk but let's say after lunch or after some time when I get tired, I just shift to my bed, especially when it's cold*' [P8]).

### 'Once I get in the zone, I can study for hours': Sitting as integral to the optimal studying mindset

>Maintaining focus on studying was often seen as challenging ('*even when I'm watching lectures I can find myself being very distracted so [it] takes a lot of willpower*' [P5]) and participants frequently spoke about needing to be in the right psychological mindset to study ('*if in the morning I'm very zoned out, I can't concentrate, I normally find it pretty pointless to even bother because I'm not going to get anything done*' [P12]). The idea of being either 'in' or 'out' of the optimal study mindset–or the 'zone'–was clearly expressed by many participants ('*I think once I get in the zone, I can go for hours on end*' [P8]). Participants often made the distinction between easier and more difficult tasks, explaining that for easier tasks 'entering' the 'zone' is not as important.

Furthermore, for many participants, the optimal study mindset could be elusive and therefore, when they 'found' it, they sought to capitalise on this by being productive as possible ('*once I do get into a mode where I can focus really well . . . I'm just like just get as much work as you can done*' [P12]). Participants reported that taking a break typically took them out of the 'zone', and so would forgo taking breaks when they were 'in the zone':

> *I think once I feel productive I don't want to leave that zone because I've got such a good momentum going and [taking breaks] sort of hinders me . . . 'cause what's the point if I'm super productive at that point?* (P8).

For some participants, even if they had the intention of taking breaks during study time, they would forget to do so once they were immersed in the zone ('*I tried to take breaks but then I forget because when I'm in the moment of studying I want to be able to finish as much as I can*' [P13]). If they did remember to take a break, they would try not to exit the zone by only taking short breaks before returning to their desks, for example, or avoiding other cognitively demanding tasks during breaks:

> *I can't afford to let myself relax completely [by taking longer breaks to walk outside] 'cause once I do, I find it hard to get back in the zone again so I prefer doing things which don't take up a lot of my time and attention* (P8).

In the same way that 'getting into the zone' eliminated break-taking for the participants, losing the zone (or failing to find it) would prompt break-taking:

> *[I take breaks from sitting] essentially when I just can't focus anymore, which can happen, depending on the material and how dry the material is, it can happen really frequently or if I've gotten into the flow of the subject and I won't break at all* (P3).

Several explained that if they were not in the zone, then trying to study would be inefficient ('*[I would rather take a break] than force myself to do something that I know I either won't do it to my full extent or I just don't understand what I'm revising*' [P1]). As a result, participants frequently described how boredom or tiredness could make them realise that they were no longer in the optimal 'mode' to study, which could prompt them to take a break from sitting ('*when I'm typing and I see myself getting slower at the typing then I realised that I might be getting tired and restless so I realised that I should take a break*' [P13]).

## Discussion

Our study demonstrated that the way in which students undertake learning activities at home has apparent implications for sitting time, and their willingness to take breaks to interrupt their sitting. Whilst participants showed awareness of the potential health risks of prolonged sitting, we found that their main priority when studying at home was being productive, and they felt that interrupting their sitting would compromise their productivity. Likely as a result, participants reported engaging in extended periods of sitting. Participants spoke of the importance of creating optimal physical and mental environments for productive study. Sitting was viewed as a core component of these environments, whereas taking breaks was seen as an unwanted distraction from studying. These findings illustrate important psychological and environmental barriers to reducing home-based sedentary behaviour among students. Given the many adverse physical and mental health outcomes associated with sedentary behaviour (e.g., type 2 diabetes, all-cause mortality [2, 4, 5], depression [6]), interventions are needed to encourage student PA throughout the university experience. Our results suggest that intervention designers should consider the home environment more explicitly when aiming to reduce student sedentary behaviour. Furthermore, the acceptability of interventions to tackle sitting when studying at home may depend on the extent to which they align with students' prioritised goal of productivity.

Our participants reported that, as part of their usual home-based study activities, they engaged in extended periods of uninterrupted sitting. While it is well-documented that university students are typically highly sedentary [10, 12], to our knowledge this is the first study to highlight the importance of home-studying practices as a facilitator of sedentary behaviour. Moreover, our findings illustrate why home-based study can be conducive to sitting. Participants showed some understanding of the adverse effects of sitting but prioritised their studying goals. They reported that sedentary behaviour, while not an intentional activity, was necessary for productivity, as they felt unable to work effectively unless they were sitting down (see too [16, 18]). Practical reasons may somewhat explain this perception as studying typically involves making notes or using a computer [19], tasks that are less feasible when standing or moving without specialist equipment (e.g., sit-stand desks [32]). Additionally, however, sitting was deemed by many to be essential for creating the right psychological mindset, or 'zone', for focused studying. Participants reported striving to 'get into the zone' for productive studying and perceived breaks as an unwanted distraction that could disrupt the studying mindset. Participants viewed break-taking as acceptable only after they had already come 'out of the zone', such as upon task completion or when experiencing tiredness or boredom that prevented them from continuing to study effectively. These findings have important implications for intervention design. While calls have been made for interventions to reduce students' sedentary behaviour by raising awareness of health risks (e.g., [21]), our results suggest that this may not be sufficient to encourage break-taking, because students view sitting as instrumental to productivity. Our findings also suggest that interventions seeking to prompt break-taking at regular intervals (e.g., every 30 mins) while studying at home, regardless of the activity that people are doing, are unlikely to be acceptable to students. Studies that have trialled such interventions have demonstrated limited effectiveness in reducing sitting [33, 34]. Instead, intervention designers need to promote increasing movement in a way that aligns with students' priorities. One promising approach may be to develop methods that can identify and intervene at optimal moments between the completion of one study task and initiation of another (i.e., 'task boundaries' [16, 35]). This would better complement students' 'natural break points' in their home-study practices [36], so minimising interference with studying. Just-in-Time Adaptive Interventions, through which bespoke behaviour change prompts are delivered to

individuals to encourage sit-stand transitions at optimal moments, may be particularly promising in this regard [37].

Interestingly, some participants viewed breaks from sitting positively and reported using them to refresh their concentration and energy. This demonstrates the potential for breaks to be valued by students as a means of promoting rather than derailing productivity. Experimental evidence supports the usefulness of taking breaks from sitting for productivity, though this has tended to focus on displacing sitting with physical activity. For example, one review concluded that breaking up sitting time with physical activity may improve cognitive performance [38], and in students specifically, physically active breaks within a lecture have been found to improve attention and learning compared to no breaks and non-exercise breaks [39]. Our results develop this work by illustrating the potential for simply interrupting sitting, without engaging in physical activity, to be beneficial for productivity. This echoes findings from qualitative studies of office workers, some of whom report using standing breaks to re-energise and gain 'thinking space' from difficult work problems [26, 36]. Students may be more receptive to breaking up prolonged sitting in response to interventions that emphasise potential benefits for enhanced performance and productivity.

Limitations of the study must be acknowledged. Firstly, we recruited participants from a larger sample of students that had previously completed a survey regarding health behaviours and home-studying. It is possible that completing the earlier survey may have raised the salience of sitting time and its potential health effects, thereby influencing the views expressed in the interview data. Specifically, participants may have shown greater interest in their sitting and reflected more deeply on their sitting behaviour than they would have done had they not completed a survey prior to the interview. Asking participants to estimate their sitting time can inadvertently motivate some to plan to reduce their sitting [40]. Secondly, the sitting time reported by participants is unlikely to be accurate, as people typically do not view sitting as an action in itself, rather time spent sitting is mentally represented as part of more meaningful actions, making it difficult to accurately recall sitting behaviour specifically [18, 40]. Thirdly, our interview questions did not probe participants' experiences of specific academic activities, such as watching lectures versus writing assignments. Future work might fruitfully explore whether certain types of home-study activity are more conducive to sitting compared to others.

Our study suggests that home-study practices may be particularly conducive to sedentary behaviour, which is known to have the potential to adversely affect physical and mental health. While no participant described sitting itself as meaningful, they commonly voiced a reluctance to break up prolonged periods of sitting, as they felt that this would jeopardise their productivity. These findings call for the development of novel ways of promoting sitting reduction within students' home-study routines in a way that facilitates, or at least does not compromise, productivity.

## Supporting information

**S1 File. Interview guide.**
(DOCX)

## Acknowledgments

The authors thank Eleanora Contini, Lian Duan, Yuxuan Lai, and Zhenyang Shao for their assistance with data collection. For the purposes of open access, the author has applied a Creative Commons Attribution (CC BY) licence to any Accepted Author Manuscript version arising from this submission.

## Author Contributions

**Conceptualization:** Rebecca Upsher, Benjamin Gardner.

**Formal analysis:** Hannah Clare Wood, Benjamin Gardner.

**Funding acquisition:** Eleanor J. Dommett, Benjamin Gardner.

**Investigation:** Sanjana Prabhakar.

**Methodology:** Rebecca Upsher, Benjamin Gardner.

**Supervision:** Rebecca Upsher, Myanna Duncan, Eleanor J. Dommett, Benjamin Gardner.

**Writing – original draft:** Hannah Clare Wood.

**Writing – review & editing:** Hannah Clare Wood, Sanjana Prabhakar, Rebecca Upsher, Myanna Duncan, Eleanor J. Dommett, Benjamin Gardner.

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
