## [Decision Letter · Decision Letter 0]

23 Sep 2024

PONE-D-24-19921Understanding university students’ experiences of sitting while studying at home: A qualitative studyPLOS ONE

Dear Dr. Wood,

Thank you for submitting your manuscript to PLOS ONE. After careful consideration, we feel that it has merit but does not fully meet PLOS ONE’s publication criteria as it currently stands. Therefore, we invite you to submit a revised version of the manuscript that addresses the points raised during the review process.

**Please see my comments in the authors section, address and upload the paper again. ** Please submit your revised manuscript by Nov 07 2024 11:59PM. If you will need more time than this to complete your revisions, please reply to this message or contact the journal office at plosone@plos.org. Please include the following items when submitting your revised manuscript:A rebuttal letter that responds to each point raised by the academic editor and reviewer(s). You should upload this letter as a separate file labeled 'Response to Reviewers'.A marked-up copy of your manuscript that highlights changes made to the original version. You should upload this as a separate file labeled 'Revised Manuscript with Track Changes'.An unmarked version of your revised paper without tracked changes. You should upload this as a separate file labeled 'Manuscript'.If applicable, we recommend that you deposit your laboratory protocols in protocols.io to enhance the reproducibility of your results. Protocols.io assigns your protocol its own identifier (DOI) so that it can be cited independently in the future. For instructions see: https://journals.plos.org/plosone/s/submission-guidelines#loc-laboratory-protocols. Additionally, PLOS ONE offers an option for publishing peer-reviewed Lab Protocol articles, which describe protocols hosted on protocols.io. Read more information on sharing protocols at https://plos.org/protocols?utm_medium=editorial-email&utm_source=authorletters&utm_campaign=protocols.

We look forward to receiving your revised manuscript.

Kind regards,

Margaret Williams, Ph.D

Academic Editor

PLOS ONE

**Journal Requirements:**

This work was supported by the Economic and Social Research Council: HW is in receipt of PhD funding from the London Interdisciplinary Social Science Doctoral Training Partnership [grant number: ES/P000703/1], https://liss-dtp.ac.uk

4. We note that this data set consists of interview transcripts. Can you please confirm that all participants gave consent for interview transcript to be published?

If they DID provide consent for these transcripts to be published, please also confirm that the transcripts do not contain any potentially identifying information (or let us know if the participants consented to having their personal details published and made publicly available). We consider the following details to be identifying information:

- Names, nicknames, and initials

- Age more specific than round numbers

- GPS coordinates, physical addresses, IP addresses, email addresses

- Information in small sample sizes (e.g. 40 students from X class in X year at X university)

- Specific dates (e.g. visit dates, interview dates)

- ID numbers

Or, if the participants DID NOT provide consent for these transcripts to be published:

- Provide a de-identified version of the data or excerpts of interview responses

- Provide information regarding how these transcripts can be accessed by researchers who meet the criteria for access to confidential data, including:

a) the grounds for restriction

b) the name of the ethics committee, Institutional Review Board, or third-party organization that is imposing sharing restrictions on the data

c) a non-author, institutional point of contact that is able to field data access queries, in the interest of maintaining long-term data accessibility.

d) Any relevant data set names, URLs, DOIs, etc. that an independent researcher would need in order to request your minimal data set.

For further information on sharing data that contains sensitive participant information, please see: https://journals.plos.org/plosone/s/data-availability#loc-human-research-participant-data-and-other-sensitive-data

If there are ethical, legal, or third-party restrictions upon your dataset, you must provide all of the following details (https://journals.plos.org/plosone/s/data-availability#loc-acceptable-data-access-restrictions):

a) A complete description of the dataset

b) The nature of the restrictions upon the data (ethical, legal, or owned by a third party) and the reasoning behind them

c) The full name of the body imposing the restrictions upon your dataset (ethics committee, institution, data access committee, etc)

d) If the data are owned by a third party, confirmation of whether the authors received any special privileges in accessing the data that other researchers would not have

e) Direct, non-author contact information (preferably email) for the body imposing the restrictions upon the data, to which data access requests can be sent

**Additional Editor Comments:**

Please review the references, some with authors names and some without, apply consistent referencing as per journal guide, throughout.

kindly indicate the impact of the survey prior to the qualitative interviews and address the reasoning behind describing the study as qualitative when a quantitative method was added/utilised.

I would expect to see something about the long term problems relating to excessive sitting while studying that were mentioned, viz., diabetes mellitus, to name one, in the discussion/conclusion section. It was addressed/raised in the introduction but there was no follow up. Is there a significance to the chronic illnesses that were mentioned as being potentially problematic, and possibly due to a sedentary student life? Please add this in to complete this thread in the article.

Reviewers' comments:

Reviewer's Responses to Questions

**Comments to the Author**

1. Is the manuscript technically sound, and do the data support the conclusions?

Reviewer #1: Yes

Reviewer #2: Partly

2. Has the statistical analysis been performed appropriately and rigorously? 

Reviewer #1: Yes

Reviewer #2: I Don't Know

3. Have the authors made all data underlying the findings in their manuscript fully available?

Reviewer #1: Yes

Reviewer #2: Yes

4. Is the manuscript presented in an intelligible fashion and written in standard English?

Reviewer #1: Yes

Reviewer #2: Yes

5. Review Comments to the Author

**Reviewer #1:** The authors showed great understanding and knowledge about the subject matter. This study will help researchers and practitioners to understand the subject matter better to help develop proper techniques to help sedentary lifestyles while improving learning

**Reviewer #2:** The sample size of this research, although has a qualitative design it is small. The findings of this study cannot be generalized since there are so few participants. The present paper does not bring new insight to the knowledge of the field. I don't consider the paper suitable for publication in this Journal.

6. PLOS authors have the option to publish the peer review history of their article (what does this mean?). If published, this will include your full peer review and any attached files.

Reviewer #1: No

Reviewer #2: No

---

## [Author Response · Author response to Decision Letter 0]

3 Oct 2024

Journal Requirements: 

Our response: We have reformatted the manuscript in line with the style requirements. 

2. Please provide an amended statement that declares *all* the funding or sources of support (whether external or internal to your organization) received during this study

Our response: Revised statement: This work was supported by the Economic and Social Research Council: HW is in receipt of PhD funding from the London Interdisciplinary Social Science Doctoral Training Partnership [grant number: ES/P000703/1], https://liss-dtp.ac.uk. There was no additional external or internal funding received for this study. 

3. When completing the data availability statement of the submission form, you indicated that you will make your data available on acceptance. We strongly recommend all authors decide on a data sharing plan before acceptance, as the process can be lengthy and hold up publication timelines.

Our response: This statement is no longer relevant, as we have now uploaded the data to the Open Science Framework (https://osf.io/6rdvf/?view_only=9b901d9f4f504dd3877dc3f9dbc0848d). (The OSF link for peer review will be converted to a public page and the link included in the paper upon acceptance.)

4. We note that this data set consists of interview transcripts. Can you please confirm that all participants gave consent for interview transcript to be published?

Our response: We can confirm that all participants gave consent for their anonymised interview transcript to be uploaded to an online repository.

5. Please review your reference list to ensure that it is complete and correct.

Our response: All references have been checked and we can confirm that the list is complete and correct, so no changes were made.

Editor Comments: 

1. Please review the references, some with authors names and some without, apply consistent referencing as per journal guide, throughout.

Our response: Thank you for highlighting this, we have removed all names.

2. Kindly indicate the impact of the survey prior to the qualitative interviews and address the reasoning behind describing the study as qualitative when a quantitative method was added/utilised.

Our response: The survey that was conducted was part of a separate (as yet unpublished) study regarding the predictors of sitting time. The only data used from the survey for the current study related to participant demographics. The present study meets the definition of a ‘qualitative study’ in that only qualitative data are used to address the research question; quantitative data are used only for sample description purposes.

We have added a sentence explaining that the demographic information collected via this survey was used to describe the sample (line 98 in track changes document). We have also expanded our reflections, in the limitations section of the discussion, on the potential impact of completing a survey on the interview data (lines 410-418).

3. I would expect to see something about the long term problems relating to excessive sitting while studying that were mentioned, viz., diabetes mellitus, to name one, in the discussion/conclusion section. It was addressed/raised in the introduction but there was no follow up. Is there a significance to the chronic illnesses that were mentioned as being potentially problematic, and possibly due to a sedentary student life? Please add this in to complete this thread in the article.

Our response: Thank you for your comment, we agree this should have been picked up in the discussion. We have added two sentences in the first paragraph of the discussion (line 351 and 356-8) to highlight the point that our participants reported high levels of sitting and, given the adverse physical and mental health outcomes associated with this, interventions are needed to help students reduce their sedentary behaviour and increase their PA.

Reviewer 1 Comment: The authors showed great understanding and knowledge about the subject matter. This study will help researchers and practitioners to understand the subject matter better to help develop proper techniques to help sedentary lifestyles while improving learning.

Our response: We thank the reviewer for their positive comments.

Reviewer 2 Comment: The sample size of this research, although has a qualitative design it is small. The findings of this study cannot be generalized since there are so few participants. The present paper does not bring new insight to the knowledge of the field. I don't consider the paper suitable for publication in this Journal.

Our response: While we note the reviewer’s concerns around sample size, such concerns are less valid to qualitative research. The aim of a qualitative study like ours is to document reflections and perspectives on a topic, not to provide definitive and generalisable conclusions. See for example Smith (2018, p137), who states that “it is a misunderstanding to claim that qualitative research lacks generalizability […] statistical types of generalizability that inform quantitative research are not applicable to judge the value of qualitative research or claim that it lacks generalizability”; https://doi.org/10.1080/2159676X.2017.1393221).

---

## [Editor Report · Decision Letter 1]

18 Nov 2024

Understanding university students’ experiences of sitting while studying at home: A qualitative study

PONE-D-24-19921R1

Dear Dr. Wood

We’re pleased to inform you that your manuscript has been judged scientifically suitable for publication and will be formally accepted for publication once it meets all outstanding technical requirements.

Kind regards,

Margaret Williams, Ph.D

Academic Editor

PLOS ONE
---

## [Editor Report · Acceptance letter]

27 Nov 2024

PONE-D-24-19921R1 

PLOS ONE

Dear Dr. Wood, 

I'm pleased to inform you that your manuscript has been deemed suitable for publication in PLOS ONE. Congratulations! Your manuscript is now being handed over to our production team.

Kind regards, 

on behalf of

Professor Margaret Williams 

Academic Editor

PLOS ONE